# Reductive Transformation of O-, N-, S-Containing Aromatic Compounds under Hydrogen Transfer Conditions: Effect of the Process on the Ni-Based Catalyst

**DOI:** 10.3390/molecules28207041

**Published:** 2023-10-12

**Authors:** Nikolai S. Nesterov, Vera P. Pakharukova, Alexey A. Philippov, Igor P. Prosvirin, Anton S. Shalygin, Oleg N. Martyanov

**Affiliations:** Boreskov Institute of Catalysis SB RAS, Academician Lavrentiev Ave. 5, Novosibirsk 630090, Russiaverapakharukova@yandex.ru (V.P.P.); philippov@catalysis.ru (A.A.P.); antonchem86@mail.ru (A.S.S.); oleg@catalysis.ru (O.N.M.)

**Keywords:** transfer hydrogenation, supercritical coprecipitation, Ni-catalysts, alcohols

## Abstract

The influence of the reaction medium on the surface structure and properties of a Ni-based catalyst used for the reductive transformations of O-, N-, and S-containing aromatic substrates under hydrogen transfer conditions has been studied. The catalysts were characterized by XRD, XPS, and IR spectroscopy and TEM methods before and after the reductive reaction. It has been shown that the conversion of 1-benzothiophene causes irreversible poisoning of the catalyst surface with the formation of the Ni_2_S_3_ phase, whereas the conversion of naphthalene, 1-benzofuran, and indole does not cause any phase change of the catalyst at 250 °C. However, after the indole conversion, the catalyst surface remains enriched with N-containing compounds, which are evenly distributed over the surface.

## 1. Introduction

The hydrotreating of petroleum products is one of the largest catalytic processes [1,2]. The main purpose of hydrotreating is to remove heteroatoms such as oxygen, nitrogen, and sulfur from oil [3,4,5]. Another important hydrotreating process is the hydrogenation of unsaturated bonds including aromatic compounds [6,7].

Hydrogen gas is associated with various disadvantages for hydrotreating processes, including an explosion hazard, high corrosion activity on metals, and the need for heavy pressure vessels during transport. Currently, alternative approaches to the purification and processing of crude oil and its by-products are being researched worldwide [8,9], one of which is the use of supercritical fluids (SCF) as active reaction media for the deep transformation of the crude oil components and their targeted “refining” [10,11,12].

Supercritical (SC) alcohols are of interest as a medium for the purification and processing of petroleum feedstocks [13,14,15]. For example, the use of SC methanol for the non-catalytic processing of asphaltenes enables the production of liquid products with a yield exceeding 80% [16]. It has been shown in [17,18,19] that SC methanol can effectively reduce the content of heteroatoms (N, S) and metals (Ni, V, etc.) in heavy petroleum feedstocks. Also, non-catalytic refining of oil in the medium of SC alcohols makes it possible to increase the content of maltene [20] and the naphtha-diesel fraction [21]. It should be noted that carrying out a non-catalytic variant of the refining of petroleum raw materials using the medium of SC alcohols requires rather harsh conditions (P = 30 MPa, T = 400 °C) [22], whereas the use of heterogeneous catalysts makes it possible to soften the conditions of the refining process and make it more economically attractive [23].

Ni-based metal catalysts show high activity in the transfer hydrogenation (TH) processes of alcohol molecules for the reductive transformations of both carbonyl [24,25,26] and aromatic groups [27,28]. In addition, nickel shows high activity in the TH of phenolic compounds [29,30,31], which can serve as a model for the conversion of lignin. For example, in our recent study [32], the transformations of phenolic molecules in the medium of ethanol and 2-propanol were analyzed in detail. It is worth noting that the SC state of the substance can be used not only to carry out catalytic transformations, but also for the synthesis of the catalysts themselves [33]. For example, reduction in the SC 2-propanol environment makes it possible to obtain highly dispersed systems based on cobalt [34] and bismuth [35]. Precipitation in the SC carbon dioxide medium is also actively used for the synthesis of highly dispersed cobalt catalysts [36,37], photocatalysts based on titanium dioxide [38,39] and zinc oxide [40,41], as well as gold-based systems [42] and mixed nickel–copper solid substitution solutions [43].

Summarizing the above, it can be concluded that the hydrotreatment of oil under TH conditions using lower aliphatic alcohols can have a number of advantages associated with both the donor activity of the alcohols and the supercritical state of the reaction mixture, which allows diffusion limitations to be removed. Since hydrotreating is mainly associated with the removal of heteroatoms from oil, in this work we have studied the kinetic laws of the transformation of O-, N-, and S-containing aromatic substrates in the medium of SC 2-propanol. Also, the effect of the substrate containing a heteroatomic group on the state of the nickel catalyst obtained by precipitation in the medium of SC carbon dioxide has been deeply analyzed.

## 2. Results and Discussion

### 2.1. Reductive Transformations of O-, N-, S-Containing Aromatic Compounds under Hydrogen Transfer Conditions

Figure 1 shows the scheme of the naphthalene transformations and the quantitative composition of the reaction mixture as a function of time. The analysis of the naphthalene conversion products (see Figure 1II) shows that tetralin is the main product, while the proportion of decalin reaches only 8% with almost complete conversion of the initial substrate, which is achieved in 3 h. Apparently, the fact that the naphthalene molecule has a flat structure and therefore can reversibly be adsorbed on the catalyst surface plays a significant role in the reduction rate of naphthalene. At the same time, the adsorption of the tetralin formed during the naphthalene conversion is significantly hindered by the fact that the remaining aromatic ring has two alkyl substituents. Similar results have been obtained in other papers dealing with both transfer and conventional hydrogenation of naphthalene [44,45,46,47]. For example, it is clear that naphthalene is relatively easy to convert to tetraline, but further hydrogenation is much slower, even under H_2_.

The obtained time dependencies for the naphthalene and product contents allowed us to simulate the transformation scheme (see Figure 1I) and to calculate the rate constants for the individual stages, the results of which are presented in Table 1. For the first step, a pseudo-first order model was used for the calculations (Equation (3)). The description of the kinetic curves using such a model (dotted lines in Figure 1II) has shown that the available experimental data are in relatively poor agreement with the model used. We believe that the flat structure of an aromatic compound can lead to its more efficient adsorption on the surface of the heterogeneous catalyst, preventing the binding of other compounds. This is reflected in a low conversion rate in the initial phase of the TH and in an increase in the conversion rate as naphthalene is consumed, since a significant part of the catalyst surface becomes free for adsorption of the intermediate product of the transformation—tetralin. This effect is also observed by competitive sorption in oxidation processes [48]. As shown in Figure 1II (solid lines), the model described by Equation (4) more accurately depicts the experimental relation between the substance concentrations and time.
(1)$$\frac{{dC}_{i}}{dt}=-k_{i}\times C_{i}$$
(2)$$\frac{{dC}_{i}}{dt}=-\frac{k_{i}*C_{i}}{{\left(1+K*C(naphthalene)\right)}^{2}}$$

The calculated rate constants of the naphthalene and tetraline reduction show that the first stage proceeds at a much higher rate (see Table 1). It can also be seen that the kinetic model, taking into account the influence of “strong” naphthalene adsorption, leads to values of rate constants that are about four times higher than the values of the other model, and the constants of the tetralin conversion rate differ by about five times. From our point of view, the result obtained for two different kinetic models seems quite logical, since the “strong” adsorption of naphthalene reduces the effective concentration of the molecules on the surface of the catalyst involved in the reduction processes. The value of the constant K, which characterizes the adsorption of naphthalene, was also calculated from the kinetic data using Equation (2) by minimizing the standard deviation.

In contrast to naphthalene, the conversion of 1-benzofuran reaches 100% within 20 min after the target temperature is reached (see Figure 2II). The product analysis makes it possible to follow the successive transformations, e.g., it is clear that 2,3-dihydro-1-benzofuran, 2-ethylphenol, and 2-methylphenol accumulate first in the mixture as intermediate products. Then, they are mainly converted into 2-ethylcyclohexanol and 2-methylcyclohexanol, the contents of which in the final reaction mixture reach 73% and 15%, respectively. The formation of fully saturated perhydro-1-benzofuran (5%) and 2-ethylcyclohexanone (5%) is also observed. Between the latter product and 2-ethylcyclohexanol there is probably an equilibrium similar to that between 2-propanol on the one hand and hydrogen and acetone on the other. It can therefore be concluded that in the presence of Ni–Alum, under TH conditions, the active saturation of the double C=C bonds occurs, and splitting of one of the C-O bonds is also observed, although this does not lead to the complete removal of oxygen from the substrate molecule. The formation of 2-methylcyclohexanol indicates the cleavage of one of the carbon atoms, which is also typical of 1-benzofuran conventional hydrogenation over Ni-based catalysts [49,50].

The kinetic curves were modelled based on the scheme of 1-benzofuran transformations (see Figure 2I), while each individual reaction followed the pseudo-first order kinetics of the initial substrate (dotted lines in Figure 2II). Despite the large number of transformation products of 1-benzofuran, the modelling curve agrees well with the experimental dependencies. The values of the rate constants obtained for the oxygen-containing initial substrate (see Table 2) are significantly higher than those calculated for naphthalene. We suggest that this is due to the presence of oxygen atoms in the benzofuran molecule, as the oxygen-containing group should facilitate adsorption of the molecule onto the nickel surface. A comparison of the rate constants of the different reactions also shows that the reduction rate of the oxygenated aromatic system (k_1_) is significantly higher compared to the benzene ring (k_8_). This is an interesting result because the smaller ring in 1-benzofuran has an electron excess, but its higher reactivity can be related to a more effective interaction of the oxygen-containing group with nickel. The reduction of the aromatic rings formed in the reactions of 2-methylphenol and 2-ethylbenzene is expected to proceed at approximately the same rate (k_3_ and k_5_), whereas the rate of formation of 2-methylphenol from 2,3-dihydro-1-benzofuran (k_2_) is significantly lower than that of 2-ethylphenol (k_4_), probably due to the shortening of the carbon chain resulting in lower steric hindrances.

Unlike the previous cases, it was not possible to find a suitable transformation scheme for indole to construct a kinetic model associated with the formation of a large number of different products. Judging from the qualitative composition of the reaction mixtures, the scheme of the indole transformations is similar to that of 1-benzofurane and 2,3-dihydroindole which are formed in the first step (see Figure 3I). This is followed by cleavage of the C-N bonds, which may be accompanied by a shortening of the carbon chain. A significant difference of 1-benzofuran is that the nitrogen atoms are relatively easily alkylated by alcohol and other compounds present in the reaction mixture. Alkylation also leads to the formation of N-isopropylindole which, judging by its stable content in the reaction mixture, does not participate in further transformations. Unfortunately, using the NIST and Wiley7 databases, it is not always possible to reliably identify all of the products obtained from the hydrogenation of 2,3-dihydroindole, so in Figure 3I the numbers represent the families of probable transformation products.

Indole conversion reaches 100% within 40 min of reaching a temperature of 250 °C. The data in the literature show the negative effect of nitrogen-containing compounds on the activity of nickel-based catalysts in hydrogenation [51,52], thus negatively affecting the conversion compared to 1-benzofuran. At the same time, the adsorption of amines on the nickel surface is known to be a reversible process, and so far no significant poisoning effect has been observed at high temperatures such as 250 °C. It can be seen that in the first stage there is an accumulation of aromatic amines and imines (products 1 and 2, see Figure 3I), the concentration of which reaches a maximum during the first 20–40 min (see Figure 3II). The proportions of these products then begin to decrease, while saturated amines accumulate in the mixture (products 3). A small amount of N-isopropyl indole was also detected in the final reaction mixture, the concentration of which reached 3–4% 20 min after the start of the experiment and then remained almost constant (see Figure 3II). This fact may indicate that the nitrogen atom plays an active role in the adsorption of indole on the surface of the catalyst, since during its alkylation the rate of the reduction processes is significantly reduced.

Sulfur atoms, as well as nitrogen, are known for their toxicity in relation to nickel catalysts [53], and if for indole it was possible to achieve 100% conversion in supercritical 2-propanol, for 1-benzothiophene its value reached only 8% in 3 h (see Figure 4II). Two products were found in the final reaction mixture—2,3-dihydrobenzothiophene (4%) and ethylbenzene (4%), which are formed along parallel pathways (see Figure 4I). The formation of ethylbenzene also indicates the breaking of both C-S bonds (see Figure 4II). These results are in good agreement with the known literature data demonstrating the effective binding of sulfur-containing organic molecules to metallic nickel, leading to its deactivation [54]. It is therefore likely that the sulfur released during the formation of ethylbenzene leads to deactivation of the catalyst and the rate of reduction transformations is significantly reduced.

Thus, the hydrogen transfer processes in the 2-PrOH medium, catalysed by a Ni-containing catalyst, allow for the successful reducing transformations of both polyaromatic hydrocarbons, including O- and N-containing aromatic compounds. An attempt to restore the S-containing substrate under TH conditions does not lead to a high conversion, which is probably a consequence of the deactivation of the Ni-containing catalyst, which will be discussed in detail below.

### 2.2. Effect of Catalytic Transformations on the Structure and Composition of the Ni-Containing Catalyst

The above results show different behaviours of naphthalene, 1-benzofuran, indole, and 1-benzothiophene under Ni-catalysed TH. On the one hand, this can be explained by the different reactivity of the organic substances used, on the other hand, the interaction between the organic molecules and the catalyst surface can also play an important role. To demonstrate the latter, the catalyst samples were thoroughly characterized before and after TH.

Figure 5 shows the XRD data for the newly reduced Ni–Alum catalyst and for the catalysts after the reduction transformations of the selected substrates. Table 3 presents data on the phase composition and structural characteristics of the crystalline phases found in the catalysts after the TH. As can be seen, the samples of the catalysts before and after their use in reactions with naphthalene, 1-benzofurane, and indole are practically indistinguishable in terms of their phase composition and structural characteristics (see Figure 5a–d). Peaks of the metallic nickel phase are observed in these samples (PDF No. 04-0850) with a crystal phase size of 5–6 nm and crystal lattice parameters close to the standard value of 3.524 Å (see Table 3). In addition, the strongly broadened peaks characteristic of γ-Al_2_O_3_ (PDF No. 00-029-0063) with a spinel-type structure are observed in all samples. A significant difference in the phase composition is observed in the catalyst after conversion with 1-benzothiophene (see Figure 5e). In particular, peaks of the nickel sulfide—Ni_2_S_3_ phase with crystallite sizes of 18 nm are observed in the catalyst (PDF No. 44-1418). It should also be noted that in this sample, a decrease in the crystallite size of the metallic nickel phase to 4.5 nm and an increase in the crystal lattice parameter to 3.540 Å is observed (see Table 3). This can indicate the surface of the nickel particles is covered with sulfied and also sulfur atoms diffuse into the metal, increasing the lattice parameter.

Thus, the interaction of the Ni-containing catalyst with naphthalene, 1-benzofurane, and indole does not lead to significant structural changes in the catalyst, which is also reflected in the activity of the catalyst in reducing processes under hydrogen transfer conditions. Repeated use of the catalyst with these substrates does not result in a decrease in catalytic activity. However, the interaction of the catalyst with 1-benzothiophene under TH conditions leads to the poisoning of the catalyst with the formation of a sulfide phase which no longer shows catalytic activity and further reduction transformations become impossible. Unfortunately, metallic nickel catalysts do not allow sulfur removal even under hydrogen transfer conditions where the alcohols act as hydrogen donors.

The analysis of the survey photoelectron spectra of the samples showed that peaks characteristic of oxygen and nickel are present on the surface of all the samples studied. In addition, N1s and S2 peaks are observed for samples after reductive transformations of indole and 1-benzothiophene, respectively. The binding energy values of the photoelectronic peaks of aluminum—BE(Al2p) = 74.4 eV) (see Appendix A) and oxygen—BE(O1s) = 531.3 eV) are characteristic for these elements in the composition of the support—Al_2_O_3_ [55]. The analysis of the Ni2p photoelectronic peaks (see Appendix A) showed that nickel is present both in the form of Ni^0^ (BE = 852.7 ± 0.1 eV) and in the form of Ni^2+^ (BE = 856.1 ± 0.1 eV), which is also characterised by the presence of an intense satellite (BE = 861–862 eV) typical of this state [55].

According to the XPS method, the presence of nitrogen and sulfur can be detected on the surface of the Ni–Alum catalyst samples after the hydrogenation of indole and 1-benzothiophene in the SC 2-PrOH medium (see Table 4). The binding energy value of peak N1s (BE = 399.1 eV) (see Appendix A) is typical for nitrogen in the amino group (-NH-) [56,57,58]. The binding energy of the S2p peak (see Appendix A) is −161.9 eV and is characteristic of sulfur in the composition of metal sulfides, in this case nickel sulfide [59,60,61]. It should be noted that the sulfur content on the surface of the Ni–Alum catalyst after the hydrogenation of benzothiophene is 5.5 at.%, which is significantly higher than 1.4 at.% of the nitrogen content after the hydrogenation of indole. Thus, there is a strong interaction of the catalyst with sulfur under the conditions of the hydrogenation process in the medium of SC 2-PrOH, which is also confirmed by the presence of a low content of ethylbenzene in the products of the 1-benzothiophene conversion.

Since the XPS data show that the Ni–Alum catalyst sample has nitrogen on the surface after the reduction transformations of indole, and XRD data show that this sample does not differ significantly from the fresh catalyst. The sample was examined microscopically using elemental mapping to obtain information about the distribution of nitrogen. Figure 6 shows the microscopic data and elemental maps of the Ni–Alum sample after indole TH. As can be seen, the catalyst sample consists of spherical porous agglomerates with sizes ranging from 100 to 400 nm (see Figure 6a). These agglomerates are composed of crystallites of metallic nickel and aluminum oxide. The mapping data also show that the distribution of nickel and aluminum elements is very uniform over the volume of the agglomerates (see Figure 6b,d). It should be noted that the same tendency is observed for nitrogen. According to the elemental composition (see Table 5), the sample contains about 0.2 elemental percent of nitrogen, which is significantly less than that detected by the XPS method. This suggests that the nitrogen is probably only on the surface of the particles and not embedded in the structure of the catalyst, which is also confirmed by the XRD data (see Figure 5d). However, the nitrogen compounds are quite strongly bound to the surface of the catalyst, as they cannot be removed by washing the catalyst with a pure solvent.

To obtain more information about the adsorption phenomenon, additional IR studies were performed. The spectra of indole (a), benzofuran (b), naphthalene (c), and benzothiophene (d) solutions in 2-PrOH and pure of 2-PrOH were recorded in the ATR mode (see Figure 7). The peak at 816 cm^−1^ is characteristic of isopropanol. The peaks assigned to the CH out-of-plane bending modes of the aromatic compounds are most prominent in the 720–800 cm^−1^ region. After recording, one drop of Ni–Alum suspension in 2-PrOH was added to the above solutions. The addition of one drop of Ni–Alum suspension to the indole, benzofuran, and naphthalene solution in 2-PrOH did not cause the decrease in the peaks in the 720–800 cm^−1^ region. The observed slight decrease in peaks is associated with the dilution of the solution when adding a drop of the suspension. At the same time, the peaks of benzothiophene were decreased greatly in intensity (by six times) when the catalyst was added. This is precisely due to the “strong” adsorption of sulfur-containing compounds by the catalyst. For indole, there is no strong adsorption on the catalyst, although nitrogen was detected on the surface of the catalyst after the reaction. It is likely that the indole conversion products have a stronger adsorption and it is the presence of these adsorbed products that has been detected by XPS and microscopy methods.

It has been shown that the reduction transformations of N- and S-containing aromatic compounds in a 2-PrOH medium lead to significant changes in the composition of the Ni-containing catalyst. Thus, after the conversion of 1-benzothiophene, irreversible poisoning of the catalyst occurs with the formation of the Ni_2_S_3_ phase, whereas after the conversion of indole, an increased nitrogen content is observed only on the surface of the catalyst.

## 3. Materials and Methods

### 3.1. Materials

The following compounds were used: nickel acetate (≥99%, Across Organics, Geel, Belgium, Ni(OAc)_2_ × 4H_2_O), acetylacetone (99+%, Vecton, acac), aluminum isobutoxide (97%, Across Organic, Al(Oi-Bu)_3_), methanol (HPLC Gradient Grade, J.T. Barker, Phillipsburg, NJ, USA), 2-propanol (hc, JSC “ECOS-1”, Orenburg, Russia), CO_2_ (99.8%, Promgazservice, Orenburg, Russia), dodecane (≥99%, Sigma-Aldrich, St. Louis, MO, USA), 1-benzofuran (99%, Sigma Aldrich), 1-benzothiophene (98%, Sigma Aldrich), indole (99%, Sigma Aldrich), and naphthalene (99%, Chemical Line, Saint Petersburg, Russia).

### 3.2. Synthesis of the Ni–Alum Catalyst

The synthesis of the Ni–Alum catalyst containing 50 wt. % of the metal was carried out using a special complex SAS-50 (Waters Co., Milford, MA, USA). A methanol solution containing nickel acetate and oxide sol was injected into a stream of supercritical carbon dioxide. This led to a decrease in the solvent power of the carbon dioxide-methanol mixture, and precipitation occurred. After that, pure CO_2_ was passing through the obtained powder for 20 min to remove a residual solvent. The experimental parameters were as follows: a CO_2_ flow was 80 g/min, a solution flow was 2 mL/min, temperature was 40 °C, a nozzle diameter was 0.004″ (0.1016 mm), and a pressure was 150 bar. The concentration of sol oxide was chosen to obtain the samples with 50 wt. % oxide in the reduced metal catalyst. More information on the synthesis technique and textural characteristics of the catalyst can be found in [62].

### 3.3. Description of the Characteristics of the Ni–Alum Catalyst

Powder diffraction patterns were obtained using MoKα radiation (λ = 0.7093 Å) on a STOE STADI MP diffractometer (STOE, Darmstadt, Germany) equipped with a one-dimensional silicon strip MYTHEN2 1K detector (Dectris AG, Baden, Switzerland). Measurements were performed by scanning in the range of angles 1.5–127° in steps of 0.0015° by 2θ and the accumulation time at the point—3 s. Phase analysis of the samples was performed using the ICDD PDF database (Powder Diffraction File database PDF-2, International Centre for Diffraction Data, Newtown Square, PA, USA, 2009). The TOPAS 4.2 software package (Bruker AXS, 2009) was used to perform a full profile analysis of the powder diffraction patterns by the Rietveld method and to determine the average crystallite sizes (D_XRD_) using the Selyakov–Scherrer equation. The instrumental broadening of the diffraction lines was recorded according to the diffraction pattern of the NIST SRM 660c standard (LaB_6_). The error in determining the crystallite size was 5–10%.

The morphology and microstructure of the samples were studied by high-resolution transmission electron microscopy (HRTEM). The images were obtained using a Themis Z electron microscope (Thermo Fisher Scientific, Bleiswijk, The Netherlands) equipped with a Ceta 16 CCD sensor and a spherical aberration corrector, providing a maximum lattice resolution of 0.07 nm at an accelerating voltage of 200 kV. The samples for the HRTEM study were deposited on a perforated carbon film mounted on an aluminum grid by ultrasonic dispersion of the catalyst suspension in ethanol. Interplanar distances were calculated from Fast Fourier Transform (FFT) patterns using Velox software (Version 2020, Thermo Fisher Scientific, Waltham, MA, USA) and Digital Micrograph (Gatan, Pleasanton, CA, USA).

X-ray photoelectron spectra were measured on a SPECS spectrometer with a PHOIBOS-150-MCD-9 analyzer (AlKα radiation, hν = 1486.6 eV, 150 W). The binding energy (BE) scale was pre-calibrated using the peak positions of the Au4f_7/2_ (BE = 84.0 eV) and Cu2p_3/2_ (BE = 932.67 eV) nuclear levels. The peak binding energy (BE) was calibrated using the position of the C1s peak (284.8 eV), corresponding to the hydrocarbon-like surface deposits [55]. The sample in powder form was loaded onto a conductive double-sided copper scotch. The overview spectrum and the narrow spectra were recorded at the analyzer pass energy of 20 eV. The atomic ratios of the elements were calculated from the integral photoelectron peak intensities corrected by appropriate sensitivity factors based on Scofield photoionization cross-sections [63]. Analysis of the XPS data was performed using XPS Peak 4.1 software.

ATR-FTIR spectroscopic analysis was performed using a Bruker Vertex70v spectrometer equipped with a diamond ATR accessory (Specac Ltd., Orpington, UK) and a mercury–cadmium–telluride detector. A total of 64 scans were taken for each sample, recorded from 4000 to 500 cm^−1^ with a resolution of 2 cm^−1^. The spectra were processed using OPUS 8.5 software (Bruker Optics, Leipzig, Germany). Four solutions containing 10% by weight of 1-benzofuran or indole, 1-benzothiophene, or naphthalene in 2-PrOH were prepared. About 50 µL of each solution was placed in the special cell mounted on working surface of the diamond. After recording the spectrum, a drop of Ni–Alum suspension in 2-PrOH (contains 90 wt. % of the catalyst) was added to the same cell and the spectrum generated was recorded.

### 3.4. Catalytic Tests

To activate the catalysts, the samples (0.11–0.12 g) were reduced in a stream of H_2_ (30 L/h) for 45 min. The process was carried out in a quartz reactor at a temperature of 650 °C. After reduction, the reactor containing the sample was removed from the furnace and cooled to room temperature in the H_2_ stream. The reactor was then purged with Ar and, avoiding contact with air, placed in 30 mL of 2-propanol, which was used as a solvent and hydrogen donor.

The catalytic experiments were carried out in a batch reactor (AISI 316 L, 285 mL) at 250 °C. The autoclave was equipped with a sampling system to obtain kinetic dependencies. To prevent oxidation of the metallic nickel, the reactor was purged with argon and then a catalyst suspension in 2-propanol was added. A solution containing a substrate, 0.30 g of dodecane (internal standard) and 80 mL of 2-propanol was then added to the reactor. The substrates used were 1-benzofuran, naphthalene, indole, and 1-benzothiophene. The molar ratio of the starting substrate to nickel was approximately 10 to 1. After the catalytic experiments, the catalyst was separated from the reaction mixture, washed with 2-propanol (100 mL) and dried in air.

The following equations were used to calculate the conversion rate and the percentage composition of the mixture:(3)$${Converion}_{i}=\frac{{C\left(substrate\right)}_{start}-{C\left(substrate\right)}_{i}}{{C\left(substrate\right)}_{start}}\times 100\%$$
(4)$${Yield(product/substrate)}_{i}=\frac{{C(product/substrate)}_{i}}{\sum {C(product/substrate)}_{i}}\times 100\%$$
where C(substrate)_start_ is the initial concentration of the substrate, and C(product/substrate)_i_ is the concentration of the product or substrate at a given time.

### 3.5. Product Analysis

The liquid products of chemical transformations were analyzed using a Shimadzu GCMS-QP2010 SE chromatograph-mass spectrometer equipped with an autosampler. Capillary chromatography column DM-35 (bonded and cross-linked 35% diphenyl/65% dimethylsiloxane polysiloxane) was used (length 30 m; inner diameter 0.32 mm; fixed phase thickness 0.25 micron). Chromatograms are shown in Appendix A. The column conditioning temperature regime was as follows: 40 °C for 1 min, programmed heating to 290 °C at a rate of 15 °C per minute and 1 min at 290 °C. The evaporator temperature was 270 °C, split 1:30; helium was used as the carrier gas. The products were identified by the retention time and mass spectrum of the substance, which were compared with the corresponding data of pure compounds or with data from the electronic mass spectral libraries of NIST and Wiley7. The conversion of the starting substrates and the yield of the products were evaluated using the internal standard method, in which dodecane was used as the internal standard.

### 3.6. Calculations of Kinetic Data

The rate constants describing the transformations of the starting and intermediate compounds were calculated according to Equation (1): $\frac{{dC}_{i}}{dt}=-k_{i}\times C_{i}$, where C_i_ is the concentration of the starting or intermediate compound in solution at a given time, and k_i_ is the kinetic constant of the transformation of the starting or intermediate compound. The value of rate constants were determined by the least-squares method using the obtained analytical solution. The following function was minimized:(5)$$F\left(\vec{k}\right)=\sum_{i,j}{\left(c_{ij}^{exp}-c_{i}^{calc}(t_{j},\vec{k})\right)}^{2}$$
where $\vec{k}$ is the vector of reaction rate constants, $\vec{k}$ = (k_1_, k_2_, …); j is the serial number of an experimental point, t_j_ is the time of a j-th point, _$c_{ij}^{exp}$ is the experimental molar fraction of an i-th component at time t_j_, and $c_{i}^{calc}(t_{j},\vec{k})$ is the calculated molar fraction of the i-th component at time t_j_. The $c_{i}^{calc}(t_{j},\vec{k})$ concentrations were calculated using the Runge–Kutta method.

To calculate the conversion of naphthalene, Equation (2) was used in addition to Equation (1), which takes into account the “strong” adsorption of naphthalene on the catalyst: $\frac{{dC}_{i}}{dt}=-\frac{k_{i}*C_{i}}{{\left(1+K*C(naphthalene)\right)}^{2}}$, where K is a constant describing the equilibrium adsorption of naphthalene on the catalyst. The number of significant digits after the decimal point was determined by the minimum error value.

## 4. Conclusions

In this paper, the kinetic laws of the reductive transformations of O-, N-, and S-containing aromatic substrates under hydrogen transfer conditions are studied and determined. Additionally, we examine how these transformations impact the surface structure and properties of a Ni-based metal catalyst. It has been demonstrated that no phase change of the catalyst occurs during the reduction of naphthalene, 1-benzofuran, and indole, enabling near 100% conversion to be achieved. However, when 1-benzothiophene is converted, irreversible poisoning of the catalyst occurs with the formation of a large crystalline Ni_2_S_3_ phase, leading to a sharp decrease in the rate of the reductive transformations for this substrate.

The surface composition analysis revealed the presence of N-containing organic compounds strongly adsorbed on the catalyst surface after the indole reduction process. According to the elemental mapping data, the N-containing substances are evenly distributed over the surface of the catalyst. After the reduction transformation of 1-benzothiophene, the catalyst surface is enriched with an S-containing phase with the composition Ni_2_S_3_. During the transformations of naphthalene and 1-benzofuran, there is no significant change in the composition of the catalyst surface.

## Figures and Tables

**Figure 1 molecules-28-07041-f001:**
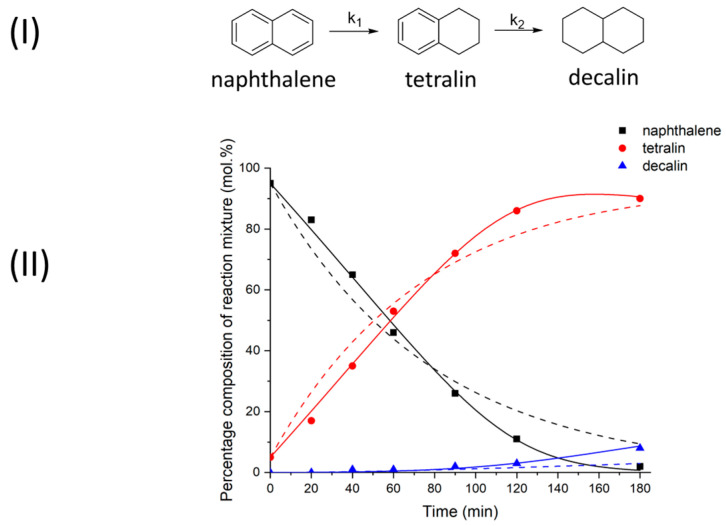
(**I**)—Scheme of catalytic transformations of naphthalene. (**II**)—Quantitative composition of naphthalene reaction products in 2-PrOH at 250 °C in the presence of Ni–Alum. The dashed lines are modeled taking into account the rate Equation (1)—$\frac{{dC}_{i}}{dt}=-k_{i}\times C_{i}$. The solid lines are modeled taking into account the “strong” adsorption of naphthalene with the rate Equation (2)—$\frac{{dC}_{i}}{dt}=-\frac{k_{i}*C_{i}}{{\left(1+K*C(naphthalene)\right)}^{2}}$.

**Figure 2 molecules-28-07041-f002:**
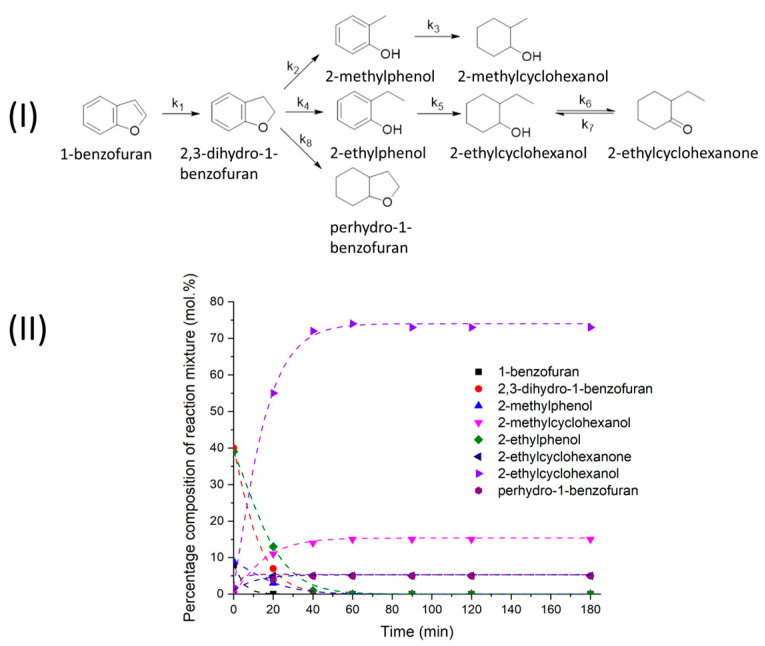
(**I**)—Scheme of catalytic transformations of 1-benzofuran. (**II**)—Quantitative composition of the 1-benzofuran reaction products in 2-PrOH at 250 °C in the presence of Ni–Alum. Dotted lines—modelling of the kinetic curves obtained using a pseudo-first order model.

**Figure 3 molecules-28-07041-f003:**
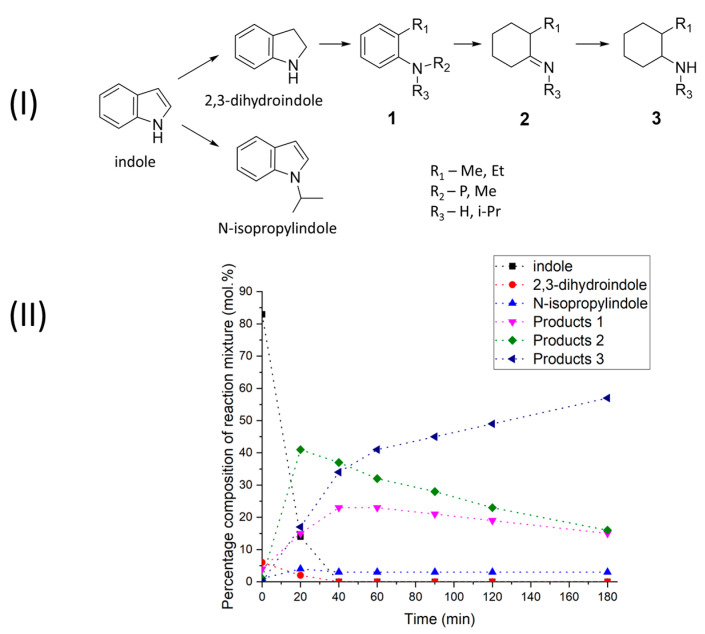
(**I**)—Scheme of the indole catalytic transformation. (**II**)—Quantitative composition of the indole reaction products in 2-PrOH at 250 °C in the presence of Ni–Alum.

**Figure 4 molecules-28-07041-f004:**
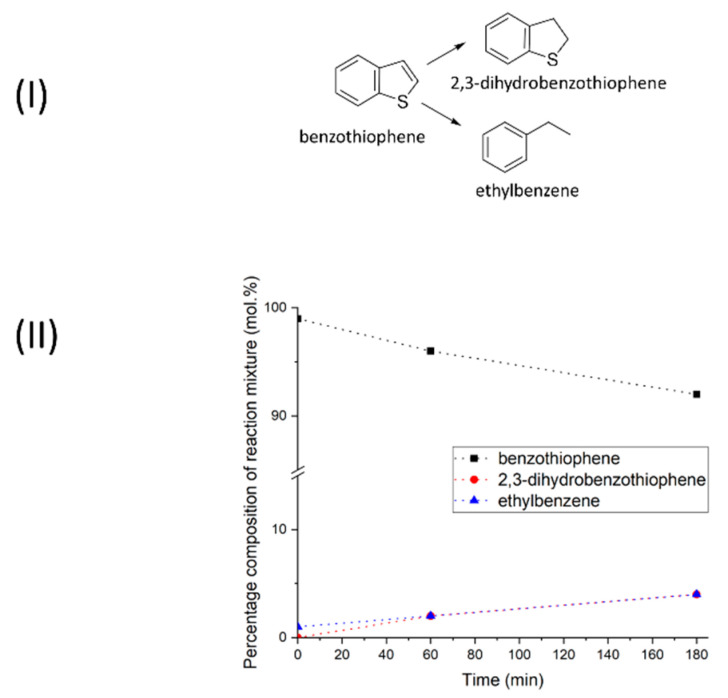
(**I**)—Scheme of the 1-benzothiophene catalytic transformation. (**II**)—Quantitative composition of the 1-benzothiophene reaction products in 2-PrOH at 250 °C in the presence of Ni–Alum.

**Figure 5 molecules-28-07041-f005:**
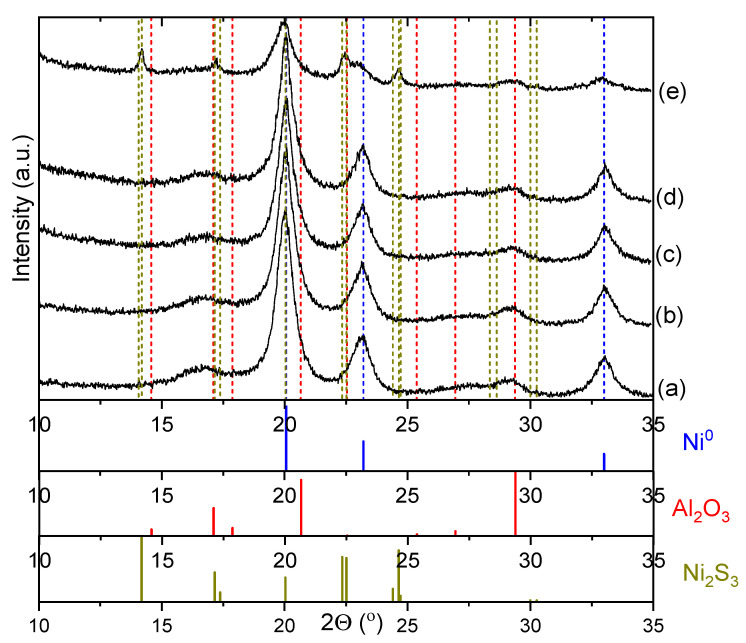
Experimental XRD patterns for Ni–Alum catalyst after transfer hydrogenation in 2-PrOH with different substrates: (a)—fresh catalyst; (b)—after naphthalene transformation; (c)—after 1-benzofuran transformation; (d)—after indole transformation; (e)—after 1-benzothiophene transformation.

**Figure 6 molecules-28-07041-f006:**
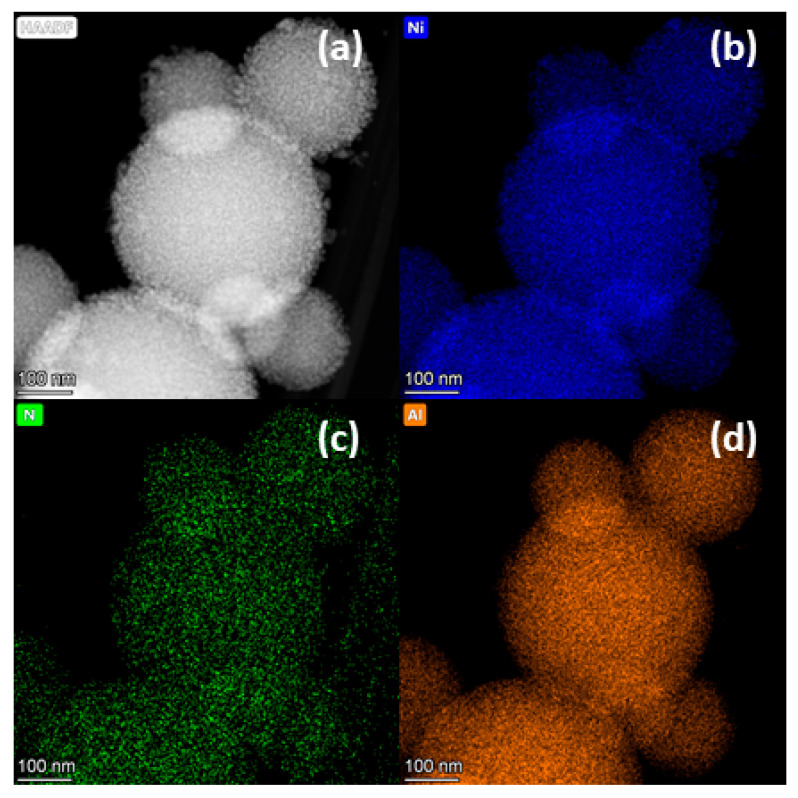
Microscopic data of Ni–Alum catalyst after indole conversion: (**a**)—High Angle Annular Dark Field (HAADF); (**b**)—nickel elemental map; (**c**)—nitrogen elemental map; (**d**)—aluminum elemental map.

**Figure 7 molecules-28-07041-f007:**
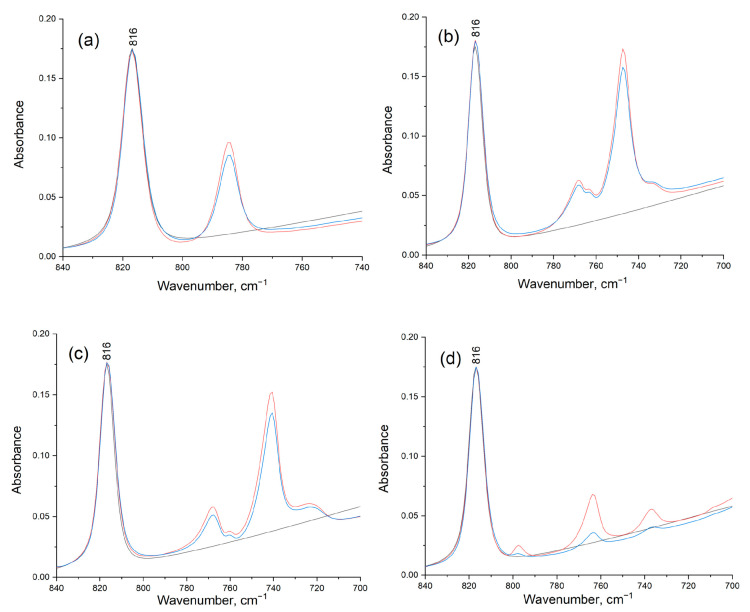
Adsorption study of naphthalene—(**a**), 1-benzofuran—(**b**), indole—(**c**), and 1-benzothiophene—(**d**). ATR-FTIR spectra of 2-PrOH—black, spectra of substrate in 2-PrOH—red, spectra of substrate in 2-PrOH with drop of Ni–Alum suspension—blue.

**Table 1 molecules-28-07041-t001:** Rate constants of naphthalene and tetrain transformations, calculated on the basis of models including (Equation (4)) and excluding the “strong” adsorption of naphthalene (Equation (3)).

Kinetic Model	k_1_ × 10^4^, s^−1^	k_2_ × 10^4^, s^−1^	K
Equation (1)	2.14 ± 0.14	0.04 ± 0.03	-
Equation (2)	8.37 ± 0.83	0.20 ± 0.02	0.016 ± 0.002

**Table 2 molecules-28-07041-t002:** Constants of the 1-benzofuran transformation rate and products calculated on the basis of a pseudo-first order kinetic model (Equation (3)).

Kinetic Model	k_1_ × 10^4^, s^−1^	k_2_ × 10^4^, s^−1^	k_3_ × 10^4^, s^−1^	k_4_ × 10^4^, s^−1^	k_5_ × 10^4^, s^−1^	k_6_ × 10^4^, s^−1^	k_7_ × 10^4^, s^−1^	k_8_ × 10^4^, s^−1^
Equation (1)	34.84 ± 22.60	2.24 ± 0.13	16.04 ± 1.37	13.24 ± 0.61	18.84 ± 0.43	20.65 ± 1.68	275.75 ± 14.86	1.51 ± 0.10

**Table 3 molecules-28-07041-t003:** XRD data of the phase composition and structural features of the crystalline phases in the catalysts before and after transfer hydrogenation in 2-PrOH with different substrates.

Sample	Phase	Lattice Parameter, Å	D_XRD_, nm
Fresh	Ni^0^	a = 3.527 (1)	5.0 (5)
	γ-Al_2_O_3_	a = 7.950 (3)	3.0 (5)
After naphthalene	Ni^0^	a = 3.528 (1)	5.5 (5)
	γ-Al_2_O_3_	a = 7.950 (3)	3.0 (5)
After 1-benzofuran	Ni^0^	a = 3.527 (1)	5.5 (5)
	γ-Al_2_O_3_	a = 7.950 (3)	3.0 (5)
After indole	Ni^0^	a = 3.527 (1)	5.5 (5)
	γ-Al_2_O_3_	a = 7.950 (3)	3.0 (5)
After 1-benzothiophene	Ni^0^	a = 3.540 (2)	4.5 (5)
	Ni_2_S_3_	a = b = 5.745 (1), c = 7.135 (1)	18.0 (5)
	γ-Al_2_O_3_	a = 7.950 (3)	3.0 (5)

**Table 4 molecules-28-07041-t004:** Surface composition and ratio of atomic concentrations of elements determined by XPS.

	Element	el.% Al	el.% O	el.% Ni	el.% N	el.% S	N/Ni	S/Ni	Al/Ni
Sample									
After naphthalene		19.0	64.9	16.2	-	-	-	-	1.2
After 1-benzofuran		18.1	63.7	18.2	-	-	-	-	1.0
After indole		20.0	64.7	14.0	1.4	-	0.10	-	1.4
After 1-benzothiophene		14.1	62.4	18.0	-	5.5	-	0.3	0.8

**Table 5 molecules-28-07041-t005:** Percentage elemental composition of the Ni–Alum catalyst after indole conversion. Data obtained from elemental mapping.

	Element	el.% N	el.% O	el.% Al	el.% Ni
Sample					
Ni–Alum catalyst after indole transformation		0.2	46.4	20.8	26.4

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
