# Peer review of "Reductive Transformation of O-, N-, S-Containing Aromatic Compounds under Hydrogen Transfer Conditions: Effect of the Process on the Ni-Based Catalyst"

_molecules, 2023, doi:10.3390/molecules28207041_

Round 1

Reviewer 1 Report

Manuscript decribes reductive transformation of aromatic compounds using Ni-based catalyst. The results are interesting and the methodology is appropriate.

Considering the obtained results, what could be the effective way to reductively transform S-containing compounds?

Could you comment on the recycling and reusing of the catalyst? Does the applied cleaning technique fully clean the catalyst? How many cycles can be performed using the same catalyst for each type of substrates? Is the catalytic efficiency the same?

Having analyzed the manuscript and data, I would recommend that this manuscript is published after minor revision.

Language quality is very good although some spelling errors occur throughout the text and should be corrected before publication.

Author Response

Manuscript decribes reductive transformation of aromatic compounds using Ni-based catalyst. The results are interesting and the methodology is appropriate.

We would like to thank the reviewer for the high appreciation of our work!

Considering the obtained results, what could be the effective way to reductively transform S-containing compounds?

The removal of sulfur from petroleum products is a targeted hydrotreating process. Unfortunately, as shown in this work, metallic nickel catalysts are not suitable for sulfur removal processes even under hydrogen transfer conditions when alcohols are used as hydrogen donors. Other systems, such as Ni-Mo containing catalysts, may be suitable for this purpose. However, this is the subject of research in our future work.

We have added additional considerations to the manuscript, Page 8, paragraph 1.

Could you comment on the recycling and reusing of the catalyst? Does the applied cleaning technique fully clean the catalyst? How many cycles can be performed using the same catalyst for each type of substrates? Is the catalytic efficiency the same?

We carried out experiments with the same catalyst and substrate twice to look at the stability of this catalyst. For indole, 1-benzofuran and naphthalene, the used catalysts give almost the same result as after the operation cycle. For 1-benzothiophene, the catalyst becomes inactive after the second experiment. It is also important to note that in this study the initial substrate/nickel mole ratio is about 10. Considering that there is only small amount of nickel is available on the catalyst surface and that the kinetics does not change during the reaction, the catalyst appears to be stable.

After washing and immersing the catalyst in a small amount of solvent (5 ml), no traces of the substrates could be detected in the solvent. So we believe this cleaning technique is appropriate.

Having analyzed the manuscript and data, I would recommend that this manuscript is published after minor revision.

Thank you very much for your estimation!

Language quality is very good although some spelling errors occur throughout the text and should be corrected before publication.

We have reviewed the text once again and corrected some spelling mistakes.

Reviewer 2 Report

The work presented by Nesterov et al. describes the synthesis the transformations of heterocyclic aromatic compounds catalyzed by Ni_Alum under hydrogen transfer conditions and the effects on the catalyst and activity after the process. Interesting data on the transfer hydrogenation processes of naphthalene, benzofuran, indole, benzothiophene and their effect on the catalytic activity and the catalyst composition have properly been commented, making the article clear for the reader. Herein only a few comments:

- representative chromatograms for the compounds studied and the intermediates detected should be reported in the supporting information.

- The conclusions should be better detailed on the effects the transfer hydrogenation processes had on the catalyst.

- Out of personal curiosity: why is necessary to activate this type of catalyst with hydrogen stream?

Therefore, this work is recommended for publication on Molecules after the Authors have modified the manuscript accordingly.

Kind regards

Author Response

The work presented by Nesterov et al. describes the synthesis the transformations of heterocyclic aromatic compounds catalyzed by Ni_Alum under hydrogen transfer conditions and the effects on the catalyst and activity after the process. Interesting data on the transfer hydrogenation processes of naphthalene, benzofuran, indole, benzothiophene and their effect on the catalytic activity and the catalyst composition have properly been commented, making the article clear for the reader. Herein only a few comments:

We would like to thank the reviewer for the comment.

- representative chromatograms for the compounds studied and the intermediates detected should be reported in the supporting information.

Thank you for your comment. We have added these chromatograms to the Supporting Information. Please, see Figures S5-S7.

- The conclusions should be better detailed on the effects the transfer hydrogenation processes had on the catalyst.

Thank you for your comment. We have added the description of the effect of all four substrates on the catalyst in conclusions: “After the reduction transformation of 1-benzothiophene, the catalyst surface is enriched with an S-containing phase with the composition Ni2S3. During the transformations of naphthalene and 1-benzofuran, there is no significant change in the composition of the catalyst surface.”.

- Out of personal curiosity: why is necessary to activate this type of catalyst with hydrogen stream?

Activation of the catalyst in the hydrogen stream is required to obtain the Ni0 metal phase. This has been demonstrated many times in the literature data that metallic nickel catalyzes the hydrogen transfer reaction more effectively compared to oxides.

Therefore, this work is recommended for publication on Molecules after the Authors have modified the manuscript accordingly.

Kind regards

Thank you very much!

Reviewer 3 Report

I think this is a great work and it can be accepted after minor remarks:

- Table 2 contains rate constants within the range from 1.5 to 275.6 s^-1. What the confidence limit for these value? Rate constants change by the more than 100 times but for all values the amount of numbers after the decimal is euqal to 2 without any comments. 

 - Authors use adsorption constant for takes into account the adsorption of naphthalene on the catalyst. How this constant was estimated? This constant is an invariant during all interval of naphthalene concentrations? I think this moment should be clarified.

Author Response

I think this is a great work and it can be accepted after minor remarks:

Thanks a lot for your comment, your opinion is very important to us!

- Table 2 contains rate constants within the range from 1.5 to 275.6 s^-1. What the confidence limit for these value? Rate constants change by the more than 100 times but for all values the amount of numbers after the decimal is euqal to 2 without any comments. 

We have added standard errors to Tables 1 and 2. To describe the number of significant digits, we were guided by the minimum error value. For Table 1, this was 0.02. Accordingly, all other values were presented with this accuracy.

 Also, we have added a comment in section 3.6.

 - Authors use adsorption constant for takes into account the adsorption of naphthalene on the catalyst. How this constant was estimated? This constant is an invariant during all interval of naphthalene concentrations? I think this moment should be clarified.

Thank you for your feedback. This constant was calculated by minimizing the standard deviation relative to the constants k1, k2, and K to solve the differential system of equations described by Equation 4 (now 2). This constant is invariant in the context of equation 4 (equation 2 in the revised manuscript).

We have added additional considerations to the manuscript, Page 3, paragraph 2.

Reviewer 4 Report

The work is dedicated, to studying the influence of the reaction environment on the surface structure and properties of materials based on a Ni-based catalyst. This catalyst is used for restoring transformations of O-, N-, S-containing aromatic substrates under hydrogen transfer conditions. The manuscript is well-written, and the experimental part allows reproducing all experiments. For the study of the catalyst, modern methods such as XPS, XRD, and TEM were utilized, indicating the reliability of the acquired data. Nevertheless, a key issue with this work is the lack of adsorption studies, even though the authors repeatedly mention that adsorption plays a significant role in the investigated process. Therefore, it is recommended to conduct an adsorption study of compounds on the catalyst and compare the adsorption curves of naphthalene, 1-benzofuran, benzothiophene and indole. It is also necessary to measure the specific surface area of the catalyst. This will allow for the confirmation of the conclusions and provide a clearer understanding of the process of catalytic compound restoration. The paper also contains several remarks related to formatting. After corrections have been made, the paper can be recommended for publication in the journal.

1.       After the equation is first mentioned in the text, it should be detailed within the text.

2.       Only a period should be used as a separator for the decimal part in Table 1.

3.       In the experimental section, it is necessary to describe the exact methodology of catalyst production in more detail. After all, all the results obtained pertain specifically to the catalyst under study, and the reference includes several catalyst production methods.

Small minor typos such as "Mpa" , "1-benzotheophene" and etc

Author Response

The work is dedicated, to studying the influence of the reaction environment on the surface structure and properties of materials based on a Ni-based catalyst. This catalyst is used for restoring transformations of O-, N-, S-containing aromatic substrates under hydrogen transfer conditions. The manuscript is well-written, and the experimental part allows reproducing all experiments. For the study of the catalyst, modern methods such as XPS, XRD, and TEM were utilized, indicating the reliability of the acquired data. Nevertheless, a key issue with this work is the lack of adsorption studies, even though the authors repeatedly mention that adsorption plays a significant role in the investigated process. Therefore, it is recommended to conduct an adsorption study of compounds on the catalyst and compare the adsorption curves of naphthalene, 1-benzofuran, benzothiophene and indole. It is also necessary to measure the specific surface area of the catalyst. This will allow for the confirmation of the conclusions and provide a clearer understanding of the process of catalytic compound restoration. The paper also contains several remarks related to formatting. After corrections have been made, the paper can be recommended for publication in the journal.

Thank you for your comment, we find it very useful to improve our work. The process of adsorption of these substrates is very important to understand the effect of the substrates on the catalyst. A significant amount of catalyst is required to conduct a detailed study of the effect of substrate adsorption. It should also be noted that the study of adsorption at room temperature does not provide information about adsorption at a process temperature of 250 °C, and it is no longer possible to study adsorption during the reaction. Following to the recommendation of the reviewer we performed an additional qualitative study of the adsorption of substrates at room temperature using FTIR spectroscopy (see Figure 7). This study showed that a strong adsorption of 1-benzothiophene is observed at room temperature, while the other substrates are practically not adsorbed on the catalyst.

The textural properties of the catalyst were studied in detail in paper 44.

After the equation is first mentioned in the text, it should be detailed within the text.

Thank you for your careful reading, we have corrected this remark.

Only a period should be used as a separator for the decimal part in Table 1.

We have corrected this comment.

In the experimental section, it is necessary to describe the exact methodology of catalyst production in more detail. After all, all the results obtained pertain specifically to the catalyst under study, and the reference includes several catalyst production methods.

We have added a description of the synthesis of the catalyst, please, see Subchapter 3.1 in the revised manuscript.

Small minor typos such as "Mpa" , "1-benzotheophene" and etc

Thank you again. We have corrected these mistakes.

Round 2

Reviewer 4 Report

The authors have made all the necessary changes. The manuscript can now be recommended for publication